# Chemical Structure Diversity and Extensive Biological Functions of Specialized Metabolites in Rice

**DOI:** 10.3390/ijms242317053

**Published:** 2023-12-02

**Authors:** Huiwen Zhou, Jinjin Zhang, Liping Bai, Jiayi Liu, Hongdi Li, Juan Hua, Shihong Luo

**Affiliations:** Research Center of Protection and Utilization of Plant Resources, College of Bioscience and Biotechnology, Shenyang Agricultural University, Shenyang 110866, Chinajiayiliu9705@163.com (J.L.);

**Keywords:** rice, specialized metabolites, chemical structure, biological function

## Abstract

Rice (*Oryza sativa* L.) is thought to have been domesticated many times independently in China and India, and many modern cultivars are available. All rice tissues are rich in specialized metabolites (SPMs). To date, a total of 181 terpenoids, 199 phenolics, 41 alkaloids, and 26 other types of compounds have been detected in rice. Some volatile sesquiterpenoids released by rice are known to attract the natural enemies of rice herbivores, and play an indirect role in defense. Momilactone, phytocassane, and oryzalic acid are the most common diterpenoids found in rice, and are found at all growth stages. Indolamides, including serotonin, tryptamine, and *N*-benzoylserotonin, are the main rice alkaloids. The SPMs mainly exhibit defense functions with direct roles in resisting herbivory and pathogenic infections. In addition, phenolics are also important in indirect defense, and enhance wax deposition in leaves and promote the lignification of stems. Meanwhile, rice SPMs also have allelopathic effects and are crucial in the regulation of the relationships between different plants or between plants and microorganisms. In this study, we reviewed the various structures and functions of rice SPMs. This paper will provide useful information and methodological resources to inform the improvement of rice resistance and the promotion of the rice industry.

## 1. Introduction

Rice (*Oryza sativa* L.) is a crop plant in the Poaceae family, and originated from China and India. Indica rice and japonica rice are derived from different gene pools of the common wild ancestor *O. rufipogon*, and are believed to have undergone multiple domestications resulting in *O. sativa*, also known as Asian common wild rice [1]. *O. rufipogon* and *O. sativa* are widely distributed throughout Asia and there is no apparent reproductive barrier between them, which results in a continuous series of intermediate and mixed genotypes, guaranteeing the formation of diverse modern rice varieties [1]. Because of the easy transportation and storage of rice, together with the wide adaptability of the plant, rice is widespread and used not only as an important food crop but also as a raw material in the processing and brewing industries. At present, hybrid rice is widely distributed throughout the Asian monsoon areas and tropical rain forests.

Specialized metabolites (SPMs) enable plants to defend themselves against biological and abiotic stresses from their environment [2,3,4]. Rice contains abundant SPMs, which are able to prevent herbivory, resist infection from pathogens and plant oxidation, and regulate plant growth and development, with some also having allelopathic effects [4,5,6,7,8]. The SPMs in rice are mainly terpenoids, phenolics, and alkaloids. Different SPMs have different organ, tissue, and expression time specificities, and are specific to different biological processes.

Terpenoids can defend directly against biological stresses. Stress induces the accumulation of momilactones, oryzalexins, and phytocassanes, and initiates the defenses mediated by these chemicals against fungi and herbivores. The biosynthesis of oryzalexins and phytocassanes from *ent*-kaurene and *ent*-cassadiene is catalyzed by CYP701A6/8 [9,10]. High concentrations of momilactones and phytocassanes accumulate at the edges of necrotic diseased leaves, preventing the subsequent spread of fungi from the infected site [11,12]. Phenolics are mainly found in the bran and husks of rice, and have direct or indirect anti-insect and anti-bacterial defense functions in addition to their antioxidant effects. Meanwhile, the momilactones and ferulic acid secreted from the roots into the soil have significant allelopathic effects and are able to inhibit the growth of harmful weeds in paddy fields. Alkaloids are released into different tissues of the rice plant and have anti-insect and anti-bacterial functions, and are also able to inhibit the growth of nearby plants when the rice is subjected to biological stress.

In recent years, the chemical structures and biological function of SPMs from rice have been partially reviewed, revealing the special diterpenoid phytoalexins and their metabolic pathway [13], as well as the related factors and signal pathways involved in regulating the production of rice phytoalexin [14]. The endogenous function of momilactone was found to be allelopathy via analysis of the phytoalexin biosynthetic genes, as well as *OsKSL4* and *OsCPS2* [15]. The genetic mechanisms behind the natural variation observed in rice SPMs have also been investigated [16]. With advances in analytical methods and spectroscopy, the number and variety of SPMs isolated from rice has increased. In this study, we review the structural diversity of the 439 SPMs found from rice to date, and summarize the pivotal roles of these SPMs in the interactions between rice and herbivores, microorganisms, and other plants.

## 2. Chemical Structure Diversity of SPMs in Rice

### 2.1. Chemical Structure Diversity of Terpenoids in Rice

Terpenoids are the main rice components with induced defensive functions. Monoterpenoids, sesquiterpenoids, diterpenoids, triterpenoids, and steroids are all important in rice defenses, and include both volatile and non-volatile terpenoids. The volatile monoterpenoids and sesquiterpenoids are often found in rice leaves. The main types of rice diterpenoids are casbene, *ent*-cassadiene, stemarene, pimarane, *ent*-gibberellin, pimaradiene, and kauranes, all of which are often involved in induced defense functions [8,17,18,19,20,21,22,23,24,25,26]. Triterpenoids are usually found in rice bran and hulls [27].

#### 2.1.1. Chemical Structure Diversity of Rice Monoterpenoids

Twenty-one monoterpenoids (**1**–**21**) have been found in rice to date. These volatile monoterpenoids largely comprise chain monoterpenoids (**1**–**4**) and cyclic monoterpenoids (**5**–**21**), which are found mainly in rice leaves, bran, seeds, and bran oil [28,29,30,31,32,33,34,35,36]. These are the volatile components responsible for most of rice’s aroma. Many volatile monoterpenoids, including geraniol (**4**) and γ-terpinene (**9**), accumulated in four-leaf-stage rice following the addition of exogenous 100 μM jasmonic acid (JA) [34].

#### 2.1.2. Chemical Structure Diversity of Rice Sesquiterpenoids

Twenty-eight sesquiterpenoids (**22**–**49**) have been isolated from rice, and are also major volatile components of rice’s aroma [28,29,30,36,37,38,39]. Sesquiterpenoids are found in all parts of the rice plant, including the leaves, husks, bran, seeds, stems, coleoptiles, roots, and rhizosphere exudates. The chemical skeletons of these sesquiterpenoid SPMs include irregular acyclic carbon (**22**–**24**), elemane (**25** and **26**), bisabolane (**28**–**31**), caryophyllane (**32**–**35**), undecane (**36** and **44**), eudesmane (**37** and **38**), muurolane (**39** and **40**), cadalenic (**41**–**43**), aromadendrane (**44** and **45**), and guaiane (**47**) [28,30,36,37,38].

#### 2.1.3. Chemical Structure Diversity of Rice Diterpenoids

Fifty-two diterpenoids (**50**–**101**) have been reported from rice [8,17,18,19,20,21,22,23,24,25,26,37,40,41,42,43,44,45,46,47,48]. Seven kinds of diterpenoid skeleton have been found in these diterpenoids: casbene (**52**–**54**), *ent*-cassadiene (**56**–**61**), stemarene (**62**–**64**), pimarane (**65**–**72**), pimaradiene (**73**–**80**), *ent*-gibberellin (**89**–**91**), and kauranes (**81**–**87** and **91**–**100**) [8,17,18,19,20,21,22,23,24,25,26,37,40,41,42,43,44,45,46,47,49].

Compounds **56**–**74** are mainly distributed in the leaves, stems, and roots, while compounds **56** and **57** are mainly found in the husks [18,19,20,21,22,23,43,44,45,47,48,49]. Momilactone B (**77**) is present in the shoots and roots of rice throughout the entire life cycle [50]. The concentrations of momilactone B (**77**) in the shoots and roots increases gradually with plant growth to the flowering stage, when it reaches its highest levels, about 245 and 64.1 nmol/g FW, respectively [51]. Oryzalic acids A–B (**83** and **84**) and oryzalides A–B (**86** and **87**) are found at the rice seedling, tillering, and mature stages. The total concentrations of these four compounds were the highest at the mature stage, with 37.9 μg/g FW [52]. The concentrations of compounds **86** and **87** (in 35.0 and 1.7 μg/g FW, respectively) in the leaves, stems, ears, and roots were significantly higher than those of compounds **83**–**84** [52].

The plant hormones gibberellins (GAs) are common diterpenoids in rice, and rice GAs include gibberellins A_1_/A_4_/A_19_ (**89**–**91**). Gibberellins A_1_/A_4_ (**89** and **90**) are found in the seed kernels of third-leaf-stage rice seedlings. Compound **91** is the main endogenous genetic factor in rice, and has low growth-promoting activity. The concentrations of compound **91** vary significantly throughout the plant life cycle, usually reaching the highest levels in third-leaf-stage seedlings. Moreover, the levels of compound **89**, which is involved in the regulation of plant growth and development, may be controlled by the rates of biosynthesis and metabolic transformation of gibberellin A_19_ (**91**) [46].

#### 2.1.4. Chemical Structure Diversity of Triterpenoids and Steroids from Rice

Thirty-five triterpenoids (**102**–**136**) have been isolated and identified from rice, and are mainly found in rice bran [53,54,55,56,57,58,59,60,61,62,63,64,65,66]. The major types include citrostadienol (**104**–**107**), gramisterol (**108**–**110**), cycloeucalenol (**111**–**113**), cycloartanol (**114** and **115**), tetracyclic triterpene (**118**–**120**), cycloartane (**124**–**128**), orizalanosterolide (**130**–**131**), and lupane (**132**–**133**) [53,54,55,56,57,61,63,64,65,67,68]. Compounds **104**–**113** and their derivatives are considered to be nortriterpenoids according to their biosynthetic pathways.

Forty-five steroids (**137**–**181**) have been isolated from rice, and are also usually found in rice bran [6,53,54,55,56,63,68,69,70,71,72,73]. Both hydrophobic (**138**–**142**) and ergostane (**147**–**168**) skeletons have been found [55,70,74]. The γ-oryzanol isolated from rice bran is a mixture of triterpenols and phytosterol ferulates. Cycloartanyl ferulate (**115**), 24-methylene cycloartanyl *trans*-ferulate (**119**), 24-methylene cycloartanyl *cis*-ferulate (**120**), and cycloartenyl ferulate (**128**) were found to be the main ingredients of γ-oryzanol [75]. Isolated phytosterols include campestanol (**139**), stigmasterol (**143**), and sitosterol (**154**), which are important components of the rice membrane lipid bilayer. These chemicals are able to regulate the fluidity of the membrane, and accumulate as rice seedlings mature [76].

### 2.2. Chemical Structure Diversity of Rice Phenolics

#### 2.2.1. Simple Rice Phenolics

Forty-nine simple phenolic acids (**182**–**230**) have been identified from rice [77,78,79,80,81,82,83,84,85,86,87]. The phenolics in rice exist mainly in soluble conjugated and insoluble forms, and were found covalently bound to sugar parts or cell wall structures such as cellulose, hemicellulose, lignin, pectin, and rod-like structural proteins [88]. The distribution of phenolic acids throughout the rice plant varies, but most are found in the insoluble form and are bound, and have a strong antioxidant capacity. *p*-hydroxybenzaldehyde (**183**) is a major phenolic compound in rice hulls, and other important phenolic compounds in rice include *p*-hydroxybenzoic acid (**184**), vanillic acid (**189**), caffeic acid (**190**), *trans*-ferulic acid (**194**), *cis*-ferulic acid (**197**), sinapic acid (**203**), syringic acid (**211**), and protocatechuic acid (**214**) [84]. Compounds **217** and **194**/**197** accumulate in rice bran [7,80], and compounds **194**/**197** are the most enriched phenolic acids in insoluble binding sites [89]. More than 90% of the phenolic acid and antioxidant activity in the whole of the rice plant is seen in the bran and husk. The diversity of phenolic acids and their antioxidant capacities were generally higher in pigmented rice than in non-pigmented rice [90,91].

#### 2.2.2. Rice Flavonoids

One hundred and twenty-six (**231**–**356**) flavonoids have been identified from rice, with the observed skeletons including diphenylpropane (**233**), non-prenylated flavanone (**235**–**239**), isoflavone (**335** and **336**), and anthocyanin (**337**–**354**) [69,92,93,94,95,96,97,98,99,100,101,102,103,104,105]. Compounds **334**–**354** were anthocyanins, which are widely found in pigmented rice [106,107]. Compounds **335**–**336** and **347** are commonly found in purple rice bran [87]. Naringenin (**244**) is considered to be the biosynthetic precursor of sakuranetin (**246**), and the conversion of naringenin (**244**) into sakuranetin (**246**) is catalyzed by naringenin 7-*O*-methyltransferase (OsNOMT), a key enzyme in the synthesis of sakuranetin (**246**), in rice leaves [95]. Compounds **275** and **276** have been identified from the whole rice leaves and phloem [108]. The variety *O. sativa* spp. *japonica* cv. Hwa-Young also has a high concentration of flavonoids in the seeds, especially in the endosperm tissues, and eight different flavonoids (**292**–**295**, **257**–**260**) have been isolated [109,110]. Tricin (**297**) accumulates in rice leaves, stems, and roots [111,112].

#### 2.2.3. Other Rice Phenolics

Seventeen other phenolic compounds (**356**–**372**) have been identified from rice [64,82,92,113,114,115,116,117]. These include tocopherols (**358**–**361**) and tocotrienols (**362**–**365**). Tocotrienols are unsaturated forms of vitamin E with three double bonds on the terpenoid side chain [118].

### 2.3. Chemical Structure Diversity of Rice Alkaloids

Forty one alkaloids (**373**–**413**) have been identified from rice, including indolamides (**373**–**380**), amides (**381**–**399**, **403**), and nitrogen-containing heterocyclic rings, formed by the decarboxylation of dicarboxylic acid (**401** and **404**–**413**) [82,119,120,121,122,123,124,125,126,127,128,129,130,131,132,133]. 2-acetyl-1-pyrroline (**381**), a nitrogen-containing aromatic compound, is present at high concentrations in brown rice, and gives it its unique “popcorn” flavor [124]. In fragrant rice cultivars, the synthesis of 2-acetyl-1-pyrroline (**381**) from proline is regulated by the pyrroline-5-carboxylic acid synthase (*P5CS*) gene [134].

### 2.4. Chemical Structure Diversity of Other Types of Compounds in Rice

Twenty-six other types of compounds (**414**–**439**) have been identified from rice, including chain hydrocarbons (**414**–**429**, **438**), alicyclic hydrocarbons (**430**, **432**–**437**, **439**), and aromatic hydrocarbons (**431**) [30,33,64,73,135,136,137,138,139,140,141,142,143]. Compounds **419** and **435**–**437** are found throughout the whole rice plant. Compounds **414**–**418** are generally found in rice kernels. Compounds **423**–**427** are generally found at higher concentrations in the seed shell than elsewhere in the rice plant.

## 3. Interactions between Rice SPMs and Herbivores

### 3.1. Rice SPMs Related to Herbivore Resistance

#### 3.1.1. Anti-Insect Activity of Rice Terpenoids

The expression of genes involved in terpenoid synthesis is usually elevated following herbivory, and high concentrations of terpenoids are synthesized to resist biological stress. For example, when the rice phloem is subjected to herbivory by *Sogatella furcifera*, a mixture of volatile sesquiterpenoids is obtained from the damaged parts (Table 1) [144,145,146]. These volatile sesquiterpenoids include (*E*)-γ-bisabolene (**28**) and α-zingiberene (**31**), which are some of the products of the sesquiterpene synthases encoded by the *Os08g07100* and *Os08g04500* genes. Linalool (**3**) from rice leaves is the single product produced by linalool synthase (*OsLIS*), which is encoded by the *Os02g02930* gene. Linalool (**3**) is released from the damaged parts of rice, and is the most abundant volatile emitted from *S. furcifera*-damaged rice plants (165.2 ng/plant/h) [39,145]. The concentration of momilactone A (**76**) in damaged leaves increases with the prolongation of infection time, and the amount of momilactone A (**76**) was more than 3 μg/g FW 7 days following *S. furcifera* herbivory [147]. Feeding by other herbivores also induced sesquiterpenoid emissions in rice. Following *Nilaparvata lugens* or *Chilo suppressalis* herbivory, the expression of the sesquiterpene synthase II gene (*OsTPS2*) in rice leaves increased to promote the concentrations of (*E*)-β-farnesene (**22**) and (*E*)-β-caryophyllene (**35**) [39].

Moreover, the diterpenoid biosynthetic pathway in rice is often activated following herbivore feeding. (*E*,*E*,*E*)-geranyl diphosphate (GGPP) is converted into *syn*-copalyl diphosphate (*syn*-CPP) in a reaction catalyzed by rice CPP synthase (*OsCPS4*) and *syn*-CPP synthase (*OsCYC1*). *Syn*-CPP can be further converted into 9β*H*-pimara-7,15-diene (**73**), in a reaction catalyzed by *OsKSL4*. Subsequently, the 9β*H*-pimara-7,15-diene (**73**) can be converted into 3β-hydroxy-9β*H*-pimara-7,15-dien-19,6β-olide (**74**) by either *CYP99A2* or *CYP99A3*, and compound **74** can in turn be converted into the momilactones A (**76**) or B (**77**), catalyzed by the OsMAS protein. These diterpenoids enhance the ability of rice to resist biological stresses (Figure 1) [47].

In addition, certain volatile terpenoids are able to attract the natural enemies of herbivores and therefore play an indirect role in rice defenses against herbivory. Linalool (**3**) and zingiberene (**31**) are the most abundant volatile sesquiterpenoids released after *S. furcifera* feeding, and are able to attract female parasitic *Cotesia marginiventris* wasps, which are a natural enemy of *S. furcifera* [145]. Similarly, linalool (**3**), (*E*)-β-farnesene (**22**), and (*E*)-β-caryophyllene (**34**) are the major monoterpenoids and sesquiterpenoids released by rice 10 days after *N. lugens* infestation, and serve as important signals enabling the natural enemies of rice herbivores, for example, *Anagrus nilaparvatae*, an egg parasite of *N. lugens*, to locate the infested rice [39].

Herbivory also mediates the levels of certain hormones in rice that reduce plant defense. Jasmonic acid (JA) concentrations significantly decreased after *N. lugens* infestation in the rice cultivar Rathu Heenati [148]. In order to reduce the influence of herbivores, rice has developed a variety of countermeasures. The expression of the resistance gene *Bph14* increases in the 24 h following initial *N. lugens* infestation, via interactions with the transcription factors WRKY46 and WRKY72. *Bph14* activates the salicylic acid (SA) signal pathway and increases the deposition of callose into the phloem cells [149,150]. Meanwhile, some OsWRKYs (OsWRKY62/50/104) and OsNACs (Os05g0442700, Os12g0630800/0156100, Os01g0862800) showed significantly higher expression under *S. furcifera* infection [151]. Rice wound-inducible transcription factor RERJ1 also participates in the JA-mediated stress response by physically binding OsMYC2 and can protect against herbivory by activating JA signals (Figure 1) [152].

#### 3.1.2. Anti-Insect Activity of Phenolic Substances in Rice

The biosynthetic pathways of various phenolic compounds in rice are related to resistance to herbivores. *N. lugens* herbivory on rice results in the conversation of 4-coumaroyl-CoA and malonyl-CoA into naringenin chalcone by chalcone synthase (CHS) in the rice tissues, and the naringenin chalcone can be further converted into naringenin (**244**) via the action of chalcone isomerase (CHI) [153]. Naringenin (**244**) is converted into apigenin (**242**) by CYP93G1 (OsFNSII), which is in turn used in the synthesis of luteolin (**240**) by *CYP75B4*. The reaction of luteolin (**240**) into tricin (**297**) is catalyzed by *CYP75B4*, and tricin is known to improve rice resistance to *N. lugens* [154]. Herbivory also increases the levels of JA in plants, resulting in enhanced activity of the *OsNOMT* promoter and therefore promoting the synthesis of phenolic compounds. After *S. furcifera* herbivory, *OsMYC2*-like proteins 1 and 2 (OsMYL1 and OsMYL2) act synergistically with OsMYC2 to further activate the *OsNOMT* promoter. JA signal transduction is reinforced by OsMYL1 and OsMYL2 via OsMYC2, resulting in the synthesis of sakuranetin (**246**) from naringenin (**244**) during the rice defense response (Figure 1) [155,156].

Herbivory is able to induce an increase in the levels of phenolic chemicals in rice. The concentration of sakuranetin (**246**) in rice leaves can reach 0.6 μg/g FW following herbivory [147]. Moreover, the concentrations of 4-hydroxybenzoic acid (4-HX) (**184**), ferulic acid (FER) (**194**/**197**), and *p*-coumaric (*p*-CM) (**219**) in the tissues of rice varieties resistant to *Oryzophagus oryzae* are significantly higher than those in the tissues of susceptible rice [157] (Table 1). After *S. furcifera* infected rice plants, the DEG expression of the *OsF3H* gene enhanced kaempferol (**278**), quercetin (**282**), cyanidin (**334**), and delphinidin (**342**) biosynthesis in response to the infestation [158,159]. The concentration of tricin (**297**) is markedly higher at the leaf stage than at the tiller or booting stages, with a significant negative correlation with rice injury severity [160].

These phenolic compounds are able to inhibit certain herbivorous behaviors and some have direct anti-insect effects. For example, rice plants treated with 0, 50, or 100 ppm of eriodictyol (**237**) show increased resistance to *N. lugens*. Schaftoside (**275**) and isoschaftoside (**276**) in rice also have antifeedant effects against *N. lugens* [161]. After feeding on rice leaves containing tricin (**297**) for 15 days, the weight of honeydew produced by *N. lugens* nymphs was found to be negatively correlated with the concentration of tricin (**297**). Furthermore, 500 μg/mL of tricin (**297**) can significantly inhibit the spawning and feeding behaviors of female *N. lugens* (Figure 1) [162]. Tricin (**297**) also acts as an inhibitor of *N. lugens*’s uptake of phloem sap and stimulates probing behavior to detect irritants. Tricin concentrations are negatively correlated with the duration of *N. lugens* phloem feeding and positively correlated with probing frequency [163]. *Laodelphax striatellus* adults fed on rice stems and leaves containing tricin 5-*O*-glucoside (**299**) and tricin 7-*O*-rutinoside (**300**) for 14 weeks displayed a marked increase in the frequency and duration of probing behavior [164] (Table 1). Thus, different phenolic chemicals in rice allow the plant to resist herbivores via several different mechanisms.

#### 3.1.3. Anti-Insect Activity of Alkaloids in Rice

Herbivory can stimulate increases in alkaloid levels in rice. *N. lugens* and *Mythimna loreyi* feeding both increased the accumulation of isopentylamine (**373**) [119] (Table 1). Moreover, 48 hours of *C. suppressalis* larvae herbivory resulted in concentrations of serotonin (**390**) and tryptamine (**391**) in rice leaves that were 3.5 times higher than those in uninfected leaves, and the concentrations of *N*-feruloyltryptamine (**395**) and *N*-*p*-coumaroylserotonin (**398**) in leaves subjected to herbivory were 33 times and 140 times higher than those in control leaves, respectively [128].

Alkaloids function similarly to phenolics and some can also directly inhibit herbivores. *N. lugens* fed on rice seedlings that have been immersed in 50 mg/L isopentylamine (**373**) solution have a higher mortality than those fed on non-treated seedlings [119]. Similarly, bioassays demonstrate that *N. lugens* that ingest a 15% sugar solution containing *N*-*p*-coumaroylputrescine (CouPut) (**377**) or *N*-feruloylputrescine (**378**) have a higher mortality than those on a sugar-only diet [123]. The levels of 14 benzamides or hydroxycinnamic acid amides in rice increase following herbivory by *S. furcifera.* These chemicals include *N*-feruloyltyramine (**376**), *N*-feruloylputrescine (**378**), *N*-*p*-coumaroylagmatine (**379**), and *N*-feruloylagmatine (**380**), and demonstrate feeding and oviposition inhibition in *S. furcifera* adult females [122]. Serotonin (**390**) and tryptamine (**391**) are active substances that affect herbivores’ nervous system, and their accumulation can directly affect the behavior and physiological functions of herbivores [128]. For example, high concentrations of tryptamine (**391**) demonstrated anti-oviposition activity against *Bemisia tabaci* [165], and both growth inhibition and antifeedant effects against *Malacosoma disstria* and *Manduca sexta* [166].

### 3.2. Salivary Metabolites from Herbivores Induce Defense Responses in Rice

Proteins in the salivary glands of herbivores have multiple effects on rice, and can not only induce the production of SPMs, but can also activate rice hormone pathways to affect the rice growth index. The protein N1G14 is generated in follicle A of the main salivary gland in *N. lugens*. N1G14 is secreted into rice plants during feeding, inducing the accumulation of reactive oxygen species, callose deposition, and the activation of the jasmonic acid (JA) signaling pathway [167]. Similarly, *N. lugens*-secreted mucin-like protein (NlMLP) is highly expressed in the salivate glands of *N. lugens* and is also secreted into rice during feeding. NlMLP induces rice genes encoding basic NbPR3 and NbPR4 proteins, which promote the JA signaling pathway, as well as the biosynthesis of callose in the cell wall [168]. Moreover, herbivore vitellogenin (VgN) is also able to induce the production of SPMs and a defense response in rice. VgN from *S. furcifera* or *L. striatellus* increases the levels of JA and induces the JA-Ile signaling pathway in rice [169]. *N. lugens* vitellogenin (NlVgN) from the salivary glands of *N. lugens* enters the damaged parts of the rice plant during feeding, inducing Ca^2+^ increases and H_2_O_2_ production in the rice cytoplasm. Meanwhile, NlVgN-induced JA-responsive genes, including *OsJAZ8*, *OsJAZ11*, *OsPR10a*, and the defense-related gene, *OsWRKY26*, were up-regulated, triggering the JA pathway and thus reducing the hatching rate of *N. lugens* eggs. NlVgN also induces the release of volatile substances such as α-thujene (**13**), linalool (**3**), (*E*)-β-caryophyllene (**35**), (*E*)-β-farnesene (**22**), and α-curcumene (**29**), which are attractive to the *N. lugens* egg parasite *A. nilaparvatae* (Figure 2) [169].

### 3.3. Adaptive Mechanisms of Herbivores to Rice Defense Response

The adaptation of herbivores to plant defense responses is key to their successful expansion. In susceptible wild-type rice, *N. lugens* feeding induces the rice cytochrome P450 gene *CYP71A1* encoding tryptamine 5-hydroxylase, which catalyzes the conversion of tryptamine (**391**) into serotonin (**390**) [170]. However, in rice mutants with an inactivated *CYP71A1* gene, serotonin (**390**) is not produced and instead high concentrations of SA are produced, making the plants more resistant to herbivory [171]. Herbivores are also able to adapt to the volatile terpenoids produced by rice. Indeed, both male and female rice leaf bugs (*Trigonotylus caelestialium*) are attracted to the volatile sesquiterpenoids released from flowering rice panicles, such as (*E*)-β-caryophyllene (**34**) and β-elemene (**26**) [172], suggesting that *T. caelestialium* is not disturbed by these volatile substances.

Certain herbivores are able to suppress the plant defense response via the substances released from their salivary glands. NlSEF1 protein, which has EF-hand Ca^2+^ binding activity, is highly expressed in the salivary glands of *N. lugens*, and is excreted into rice as *N. lugens* pierces the sieve tube. NlSEF1 inhibits H_2_O_2_ production and decreases the cytoplasmic Ca^2+^ levels in rice [173]. The salivary endo-β-1,4-glucanase (NlEG1), which has endoglucanase activity, is highly expressed in the salivary glands and midgut of *N. lugens*, and allows *N. lugens* to reach the phloem by degrading the cellulose in the rice cell wall, thus overcoming the cell wall defenses [174]. *N. lugens* salivary protein 7 (NlSP7) is highly sensitive to tricin (**297**) in rice, and can enter the phloem through the cell wall, where it interacts with tricin (**297**). This interaction of NlSP7 with tricin (**297**) decreases the expression of the flavonoid biosynthesis pathway marker genes CHS and CHI [175]. Another example is that during the *S. furcifera* feeding process, the LsPDI1 produced in the salivary glands is secreted into the rice cells to induce cell death (Figure 3) [176].

## 4. Interactions between SPMs and Microorganisms in Rice

### 4.1. Antipathogen Activities of Rice SPMs

Infection with plant pathogens can seriously harm rice growth and can lead to substantial losses in crop yield. Devastating fungal diseases, such as the rice blast fungus *Magnaporthe oryzae* (anamorph *Pyricularia oryzae*), *M. grisea* (Hebert) Barr (anamorph *Pyricularia grisea* Sacc.), the rice brown spot fungus *Cochliobolus miyabeanus*, and a variant of *Helminthosporium oryzae*, also known as brown spot fungus (*Bipolaris oryzae*), can cause serious declines in rice production [177]. Meanwhile, bacterial damage to rice leaves can also be serious. For example, *Xanthomonas oryzae* pv. *oryzae* (*Xoo*) is one of the most devastating bacterial diseases of rice, and is a major obstacle to improving rice yields.

#### 4.1.1. Antipathogen Activities of Rice Terpenoids

The levels of monoterpenes and diterpenes in rice tissues increase following bacterial infection. *Xoo* infection leads to the initiation of the (*E*,*E*,*E*)-geranyl diphosphate (GGPP) pathway via the activities of *OSCPS4* and *OSKSL8* and the subsequent synthesis of oryzalexin S (**62**) [45]. Various pimaranes (**65**–**72**) and kauranes (**81**–**87** and **91**–**100**) have been isolated from the leaves of *Xoo*-resistant rice varieties [21,178,179,180], indicating that *Xoo* infection induces the accumulation of terpenoids [52]. *X. campestris* pv. *oryzae* (*Xco*) also induces terpenoid production in rice. Following infection with *X. campestris* pv. *oryzae*, the concentrations of oryzalic acid A (**83**), oryzalic acid B (**84**), oryzalide A (**86**), and oryzalide B (**87**) in rice leaves are significantly higher than in those of uninfected rice strains [181]. Some monoterpenes also show significant bacteriostatic activity; for example, at concentrations of 5 mM, the (*S*)-limonene (**6**) released from rice leaves is able to significantly inhibit the growth of the pathogen *Xoo*.

Following bacterial infection, the monoterpenoid and diterpenoid biosynthetic pathways in plants are often activated. For example, isopentenyl diphosphate (IPP) is isomerized into dimethylallyl diphosphate (DMAPP) and IPP via the action of isopentenyl diphosphate isomerase (IDI). IDI is further converted into GPP, GGPP, and FPP, which are the precursors of monoterpenoids, diterpenes, and sesquiterpenes, in reactions catalyzed by GPP synthase (GPS), GGPP synthase (GGPS), and FPP synthase (FPS), respectively [182]. GPP can then be catalyzed by the terpene synthases (TPSs) *OsTPS3/19/20/24*, to synthesize linalool (**3**), geraniol (**4**), (*S*)-limonene (**6**), and γ-terpinene (**9**), respectively, which increases the resistance of rice to bacteria [32,34,36]. Meanwhile, (*E*)-β-farnesene (**22**) and (*E*)-nerolidol (**23**) are synthesized by FPP in the cytoplasm, in reactions catalyzed by the rice terpene synthase *OsTPS18* (Figure 4) [36].

Numerous monoterpenes and diterpenes show good inhibitory activity against pathogenic fungi. After infection of rice with *M. oryzae*, GGPP is cyclized by the copalyl diphosphate (CPP) synthases (CPS) *OSCPS1* and *OSCPS2* to form *ent*-CPP, and then further converted via the action of *OSKSL7* into *ent*-cassa-12,15-diene, which is in turn converted into phytocassanes A–E (**56**–**61**), in reactions catalyzed by *CYP76M7* [183]. *ent*-CPP can be converted by *OSKSL10* into *ent*-sandaraco-pimaradiene (**71**), which can then be hydroxylated by *CYP701A8* at C-3α to form 3α-hydroxy-*ent*-sandaraco-pimaradiene (**72**), and then further cyclized by *CYP76M8* at C-7β to form oryzalexin D (**68**) (Figure 4) [48,184,185]. At 230 ppm, oryzalexin D (**68**) has a 50% inhibitory effect on *M. oryzae* mycelial growth [186]. (*S*)-limonene (**6**) can inhibit spore germination in *M. oryzae* in vitro at 50, 80, and 100 mmol/L [187]. Momilactones A (**76**) and B (**77**) isolated from rice leaves infected with *M. oryzae* inhibit the elongation of 50% of *M. oryzae* embryo tubes at concentrations of 5 and 1 μg/mL, respectively [188]. Phytocassanes A–E (**56**–**61**) also significantly inhibit the spore germination of *M. oryzae* [18]. Both phytocassanes A (**56**) and F (**61**) inhibit the mycelial growth of *M. oryzae*, and have similar inhibitory activities [20]. *P. oryzae* conidia germination and growth in complete medium are significantly inhibited in vitro at concentrations of bayogenin 3-*O*-β-D-cellobioside (**135**) of 5 nM/L–10 nM/L [66]. 3β,20β-epoxy-3α-hydroxy-5α-abieta-8,11,13-trien-7-one (**55**) and 6β,19β-epoxy-3β-hydroxy-5α,9β-pimara-7,15-diene (**80**) isolated from rice husks can inhibit 88.67% of *M. grisea* conidium germination at concentrations of 120 mg/mL [41].

#### 4.1.2. Antipathogen Activities of Rice Phenolic Compounds

Rice flavonoids have antibacterial activity. Sakuranetin (**246**) shows a strong inhibitory effect on (*Xoc*) at 10 μg/mL. The inhibitory effect of sakuranetin (**246**) on *Burkholderia glumae* and *Xoo* also increases with increasing concentrations, but was not as obvious as that on *Xoc* [121].

Phenolic biosynthetic pathways play a key role in improving resistance against fungal infection. Rice *OsF3H* and *OsF3′H* induce naringenin (**244**), which in turn produces eriodictyol (**237**) and finally synthesizes quercetin (**282**). Kaempferol (**278**) can also be formed from naringenin (**244**), finally also resulting in the production of quercetin (**282**). Quercetin (**282**) significantly reduced the metabolic activity of *Candida parapsilosis sensu stricto*, *Candida orthopsilosis*, and *Candida metapsilosis* biofilms [189]. *OsF3H* and *OsF3′H* participated in the synthesis of dihydroquercetin (**238**), naringenin (**244**), and kaempferol (**278**), which contribute to the resistance of rice to *M. oryzae* (Figure 4) [161].

Different kinds of phenolics have different effects on fungal pathogens, but all phenolics show direct inhibition activity. The growth rates of *M. oryzae* decrease with increasing concentrations of eriodictyol (**237**) [161], and tricin (**297**) inhibits the spore germination of *P. oryzae* and *R. solani* at 100 μg/g [190]. The concentrations of sakuranetin (**246**) in rice seedlings increase following inoculation with *M. oryzae* spore suspension, and sakuranetin (**246**) concentrations of 0.1 mM and 0.3 mM result in the inhibition of *M. oryzae*’s mycelium growth rate by 40% and 55%, respectively [191].

#### 4.1.3. Antipathogen Activities of Rice Alkaloids

Alkaloids have direct antibacterial activity. *N*-benzoyltryptamine (**374**) and *N*-*trans*-cinnamoyltyramine (**375**) show bacteriostatic activity against *Xoo* and *Xoc* at concentrations of 10 μg/mL, while *N*-*trans*-cinnamoyltryptamine (**397**) strongly inhibits the growth of *Xoc* between 5 μg/mL and 10 μg/mL [121]. In addition, *N*-feruloyltryptamine (**395**) and *N*-*trans*-cinnamoyltryptamine (**397**) show inhibitory activity on the growth of *Xoo* in a dose-dependent manner [120].

The concentrations of alkaloids in rice leaf tissue increases following fungal infection. For example, when rice leaves are infected with *C. miyabeanus* in vivo, the accumulation of indolamides (**378** and **379)** and amides (**392**–**394**, **396**, **398**) increases [120]. The greatest accumulation following *C. miyabeanus* infection was seen in *N*-feruloylputrescine (**378**), followed by *N*-benzoylserotonin (**392**) [120]. *M. oryzae* infection also induced the accumulation of *N*-benzoyltyramine (**374**) and *N*-benzoyltryptamine (**393**) in rice leaves [17], and serotonin (**390**) accumulates in rice leaves in response to *B. oryzae* infection [192].

These alkaloids exhibit direct inhibitory effects on fungi. Serotonin (**390**), tryptamine (**391**), and *N*-benzoyltyramine (**393**) inhibit the germination of *C. miyabeanus* conidia at 300 μM, with an inhibition rate of about 25%. *N*-benzoyltryptamine (**374**) and *N*-*trans*-cinnamoyltryptamine (**397**) also inhibit *M. grisea* mycelium growth. 3-isopropyl-5-acetoxycyclohexene-2-one-1 (**439**) inhibits *M. oryzae* and *R. solani* spore germination at concentrations of 100 μg/g [190]. Moreover, alkaloids can repair functional losses of rice mutant synthesis because they have exogenous antifungal abilities. For example, the addition of serotonin (**390**) to rice mutants restores the inhibition of *B. oryzae* hyphal growth in leaves, indicating that serotonin (**390**) confers strong resistance against *B. oryzae* [193].

Alkaloids are also able to strengthen the plant cell wall, allowing it to resist fungal infection to a certain extent. The activities of aminobenzoate synthase genes (*OASA2*, *OASB1*, and *OASB2*) increased following infection of rice with *B. oryzae* [193]. Aminobenzoate synthase proteins regulate the tryptamine pathway and induce the accumulation of serotonin (**390**), tryptamine (**391**), *N*-feruloylserotonin (**394**), and *N*-*p*-coumaroylserotonin (**398**) in the cell wall after oxidative polymerization in the leaves, forming a physical barrier against fungal infection [192].

### 4.2. Main Bacterial Targets of Rice SPMs

Identifying the bacterial target of SPMs promotes understanding of the mechanisms underlying plant resistance to pathogens. Geraniol (**4**) inhibits the growth of *Xoo* by inhibiting the down-regulation of *ZipA* and *ZapE*, which are *Xoo* genes associated with cell division [32]. Another example is the protein Hpa1, which is secreted by *Xoo* via a type III secretion system (TTSS) and causes pathogenicity in rice, and which is encoded by the *hrp* gene [194]. Phenolic compounds such as *o*-coumaric acid (**217**) can inhibit at least 60% of Hpa1 activity, which reduces the pathogenicity of *Xoo* (Figure 4) [195]. Sixty-four μg /ml kaempferol (**278**) inhibited 80% of biofilm formation. Kaempferol (**278**) inhibits biofilm formation mainly by influencing the binding of the *Staphylococcus aureus* surface anchor protein to the host matrix protein, thus reducing the adhesion of *S. aureus* [196].

### 4.3. Adaptation of Fungus to Rice SPMs

Certain pathogenic fungi are able to adapt to and detoxify rice SPMs. Sakuranetin (**246**) has an inhibitory effect on plant pathogens, and shows antifungal activity significantly higher than that of naringenin (**244**) [97,98]. However, sakuranetin can be detoxified into naringenin (**244**) and sternbin (**247**) by *M. oryzae* [98]. Similarly, sakuranetin (**246**) can also be detoxified by *Rhizoctonia solani*, where it is converted into naringenin (**244**), sakuranetin 4′-*O*-β-D-xylopyranoside (**248**), and naringenin-7-*O*-β-D-xylopyranoside (**249**). The degradation products sakuranetin 4′-*O*-β-D-xylopyranoside (**248**) and naringenin-7-*O*-β-D-xylopyranoside (**249**) show no antifungal activity [99]. *M. oryzae* can convert serotonin (**390**) into 5-hydroxyindole-3-acetic acid (5HIAA) (**389**) in culture medium, which may also be part of a detoxification process (Figure 3) [192]. Understanding the mechanisms by which fungi detoxify rice SPMs is of great importance in improving the control of fungal diseases in rice, and deserves further study.

### 4.4. Interactions between Rice SPMs and Rhizosphere Microbial Communities

Various plant hormones and SPMs can promote the proliferation and aggregation of microorganisms in the rice rhizosphere. Strigolactones are potential rhizosphere-signaling molecules, and are known to increase the abundance of *Nitrosomonadaceae* and *Rhodanobacter* [197]. The cinnamic acid (0.12 mM) (**216**) and ferulic acid (0.05–0.1 mM) (**194**/**197**) exuded from rice promote cell proliferation and chemotaxis aggregation of the rhizosphere microorganism *Myxococcus xanthus*. Exogenous ferulic acid (**194**/**197**) at 53.5 mM induces *M. xanthus* growth and up-regulates the expression of chemotactic-related genes including *FrzA*, *B*, *CD*, *E*, *G*, *F*, and *Z* [198]. The rhizosphere microbial community can be regulated by plant-synthesized SPMs. The size and diversity of the rhizosphere microbial community is reduced following the inhibition of *PAL* gene expression, with only six microbial communities subsequently detected, including *Proteobacteria*, *Firmicutes*, *Spirochaetes*, *Tenericutes*, *Clostridium*, and an unknown bacteria [199]. Moreover, inoculation with endophytic bacteria induces SPM aggregation, but also promotes plant growth. Colonization of *Azospirillum* sp. B510 in rice rhizomes results in increased levels of hydroxyl cinnamon derivatives [200]. Phenolics such as gallic acid (**215**), *trans*- or *cis*-ferulic acid (**194**/**197**), and cinnamic acid (**216**) accumulate in the rice roots and leaves following inoculation with *Rhizobium leguminosarum* bv. *phaseoli* or *R. leguminosarum* bv. *trifolii*. In addition, both strains of *Rhizobium* promote the growth and productivity of rice plants under greenhouse conditions [201]. In conclusion, plants can alter the colonization of the rhizosphere by endophytic microorganisms via the release of SPMs, and endophytic microbes also affect the release of SPMs.

## 5. Rice SPMs Regulate Plant-to-Plant Relationships

Rice terpenoids and phenolics exhibit obvious allelopathic effects. They can inhibit the growth of weeds surrounding the rice plants, but also have negative effects on the rice itself.

### 5.1. Allelopathy of Rice Terpenoids

Diterpenoids, including momilactones, have obvious allelopathic effects. The momilacontes A (**76**) and B (**77**) being released by rice plants into the surrounding soil at concentrations greater than 1 μM and 10 μM, respectively, inhibit the growth of the harmful weeds *Echinochloa crus-galli* and *Echinochloa colonum* [8]. Momilaconte B (**77**) is a major allelopathic chemical in rice, and exhibits stronger weed growth inhibition activity than momilactone A (**76**). Momilactone B (**77**) inhibits 50% of the root and hypocotyl growth in seedlings of cress (*Lepidium sativum* L.) at 36 and 41 μM, and 50% of theroot and hypocotyl growth of lettuce (*Lactuca sativa* L.) seedlings at 56 and 79 μM, respectively [202]. Momilactone synthesis is deficient in rice *OsCPS4* knockdown mutants, and the levels of *cps4-tos* are reduced. When *OsCPS4* rice mutants are grown allopathically with *Lactuca sativa* seedlings, the root and hypocotyl lengths of the adjacent *L*. *sativa* seedlings increase compared to those of the control, further demonstrating the allelopathic effect of momilactones on neighboring plants (Figure 5) [203]. Triterpenes and flavonoids from rice also have allelopathic effects on neighboring plants. For example, the lanast-7,9(11)-dien-3α,15α-diol-3α-D-glucofuranoside (**116**) extracted from rice husks inhibits the growth of the duckweed *Lemna paucicostata* by decreasing its chlorophyll content [58]. Moreover, when the concentrations of β-sitosterol-β-D-glucoside (**164**) in rice husk extract reach 100 μg/mL, 10 days following the addition of the husk extract to medium containing *Microcystis aeruginosa*, the cell growth is inhibited by more than 40% [6]. However, certain triterpenoids, including momilactones, also inhibit the growth of the rice plants themselves. Momilactones A (**76**) and B (**77**) inhibit the growth of the rice roots and buds at the seedling stage at concentrations greater than 100 μM and 300 μM, respectively. This indicates that the inhibitory effect of the momilactones on the growth of rice seedlings is much lower than their effects on the surrounding weeds [8].

### 5.2. Allelopathy of Rice Phenolic Compounds

The phenolic compounds exuded from rice can also have allelopathic effects [204]. When the rice seedlings are grown surrounded by *Echinochloa crus-galli* seedlings, the allelopathic rice seedlings release lignin-related phenolic acids (*p*-hydroxybenzoic (**184**), vanillic acid (**189**), caffeic acid (**190**), ferulic acid (**194**/**197**), syringic acid (**211**), and *p*-coumaric acid (**219**)) into the soil in response [205]. *p*-coumaric acid (**219**) inhibits *E. crus-galli* roots at concentrations higher than 5 mM and also the germination of *Lactuca sativa* seedlings at 1 mM [206]. Similarly, *p*-hydroxybenzoic acid (**184**), 4-hydroxyphenylacetic acid (**186**), 2-hydroxyphenylacetic acid (**187**), vanillic acid (**189**), caffeic acid (**190**), *trans*-ferulic acid (**194**), syringic acid (**211**), cinnamic acid (**216**), *p*-coumaric acid (**219**), and salicylic acid (**226**) in rice root exudates inhibit the growth of *Sagittaria montevidensis* roots at 1000 μM [79]. The 5,4′-dihydroxy-3′,5′-dimethoxy-7-*O*-β-glucopyranosyl flavone (**298**) and 7,4′-dihydroxy-3′,5′-dimethoxy-5-*O*-β-glucopyranosyl flavone (**299**) secreted from rice roots into the rhizosphere, and converted into the aglycone form 5,7,4′-trihydroxy-3′,5′-dimethoxyflavone (**297**), show inhibitory effects on weeds such as *E. crus-galli*, as well as on microorganisms (Figure 5) [207]. Orizaanthracenol (**367**) is exuded from rice roots and can inhibit radish germination at 40 ppm, leading to reductions in total radish dry weight of 50.96%. Furthermore, the inhibitory effects of 1-hydroxy-7-((2*S*,3*R*,4*R*,5*S*)-2″,3″,4″-trihydroxy-5″-(hydroxymethyl)tetrahydro-2*H*-pyran-1-yloxy)anthracen- 2-yl 3′,7′,11′,15′,19′-pentamethyltricosanoate (**371**) on the germination and growth of radishes are 27% lower than those of 1-hydroxy-7-((2*S*,3*R*,4*R*,5*S*)-2″,3″,4″-trihydroxy-5″-(hydroxymethyl)tetrahydro-2*H*-pyran-1-yloxy)anthracen- 2-yl 3′,7′-dimethyloctanoate (**369**) [115]. In the future, the various allelopathic substances produced by different rice varieties could be used in intercropping, so as to reduce pesticide use and promote rice yields.

## 6. Endophytic Microorganisms Promote Rice Growth

Rice endophytic fungi in the rhizosphere can improve rice’s resistance to pathogens. The rice endophytic fungus *Trichoderma longibrachiatum* EF5 has indirect antagonistic activity against the plant pathogens *Sclerotium rolfsii* and *Macrophomina phaseolina* because it releases mVOCs such as α-cuprenene (**37**) [38]. Some endophytic bacteria in rice also have dual functions, both promoting rice growth and improving resistance against pathogens. For example, the rice endophytic fungus *Phomopsis liquidambaris* B3 significantly up-regulates the expression levels of *OsAOX*, *OsLOX*, *OsPAL*, and *OsPR10* in rice, improving the diversity of the microbial community in the rhizosphere and promoting rice root development. At the same time, the presence of *P. liquidambaris* B3 results in a 41.0% inhibition rate of rice spikelet disease being induced by *Fusarium proliferatum* [208]. The *Pseudomonas* strains Pf1, TDK1, and PY15 found in the rice rhizosphere promote the growth of rhizosphere microorganisms and improve the resistance of the rice plant to *Sarocladium oryzae* [209]. Similarly, rice plants treated with both *Streptomyces shenzhenensis* TKSC3 and *Streptomyces* sp. SS8S show increased activity of β-1,3-glucanase (GLU) and peroxidase (POX), which enhance the plant cell wall and promote the growth of the plant, which then demonstrates significant resistance to *Xoc* [210].

## 7. Prospect

Rice SPMs exhibit structural diversity and have extensive and varied biological activities. The diterpenoids, phenolics, and alkaloids present in rice tissues are able to reduce the growth or reproduction of certain herbivores. These chemicals can also be induced by pathogenic fungi, and not only play a direct role against pathogens but also regulate the allelopathic interactions between rice and other neighboring plants. The phenolic compounds in rice also enhance stem and leaf hardness by increasing wax deposition. Together, these SPMs play a significant role in improving rice yield and quality. In the future, rice SPMs could be used to synthesize biological herbicides and insecticides that are friendly both to humans and nature and are less dangerous than synthetic pesticides. These future herbicides and insecticides should be able to reduce harmful weeds and plant pathogens, regulate rice rhizosphere microorganisms, improve the soil and crop quality, and increase crop yields [211]. The use of the allelopathic properties of rice SPMs to manage weeds and pests would solve the problems of environmental pollution and food safety caused by traditional pesticides, and would contribute to the sustainable development of ecologically friendly agriculture [212].

## 8. Materials and Methods

Two databases, SciFinder and Web of Science, were used to investigate references from 1976 to 2023. The keywords were rice, rice-specialized metabolism, biological functions, including the biosynthetic pathway and biological functions of terpenes, phenolics, and alkaloids, as well as the targets of these specialized metabolites on herbivores, and the detoxification effects of herbivores and fungi on specialized metabolites. 212 references were cited.

## Figures and Tables

**Figure 1 ijms-24-17053-f001:**
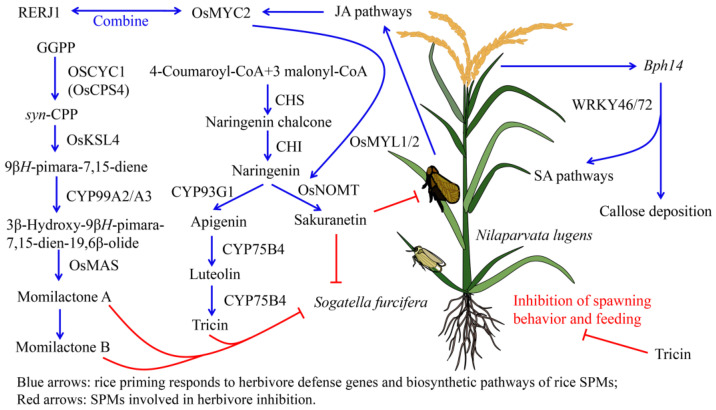
Rice SPMs with functions in plant defenses against herbivores and their biosynthesis. Blue arrows: rice priming responds to herbivore defense genes and biosynthetic pathways of rice SPMs; Red arrows: SPMs involved in herbivore inhibition.

**Figure 2 ijms-24-17053-f002:**
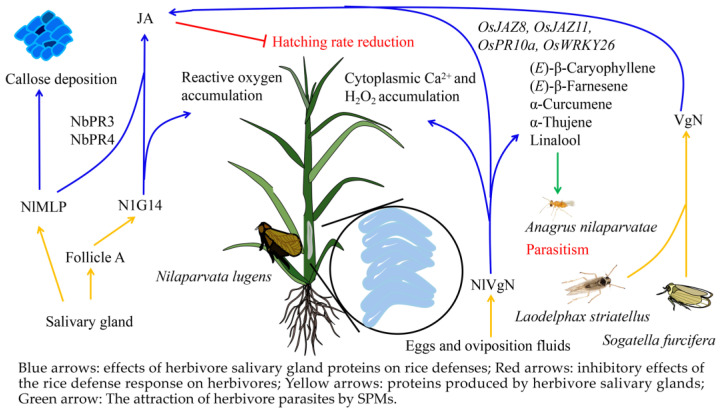
Salivary metabolites from herbivores that are known to induce defense responses in rice. Blue arrows: effects of herbivore salivary gland proteins on rice defenses; red arrows: inhibitory effects of the rice defense response on herbivores; yellow arrows: proteins produced by herbivore salivary glands; green arrow: the attraction of herbivore parasites by SPMs.

**Figure 3 ijms-24-17053-f003:**
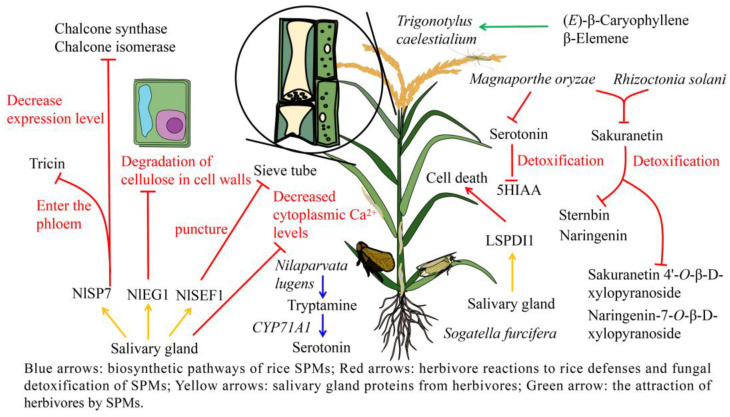
Adaptation of herbivores and fungi to rice SPMs. Blue arrows: biosynthetic pathways of rice SPMs; red arrows: herbivore reactions to rice defenses and fungal detoxification of SPMs; yellow arrows: salivary gland proteins from herbivores; green arrow: the attraction of herbivores by SPMs.

**Figure 4 ijms-24-17053-f004:**
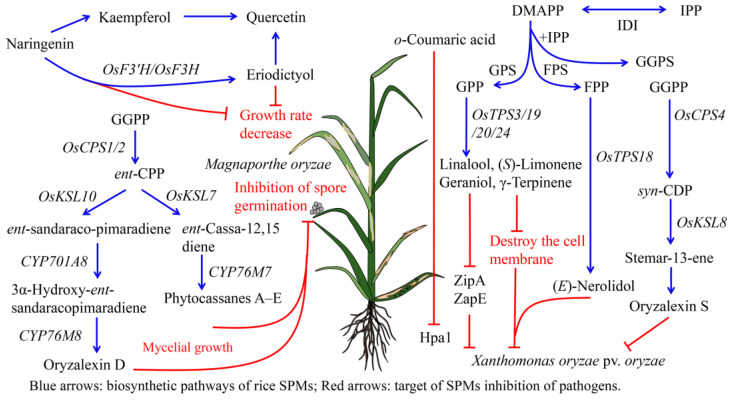
Pathogen targets of rice SPMs and the SPMs biosynthetic pathways. Blue arrows: biosynthetic pathways of rice SPMs; red arrows: target of SPMs inhibition of pathogens.

**Figure 5 ijms-24-17053-f005:**
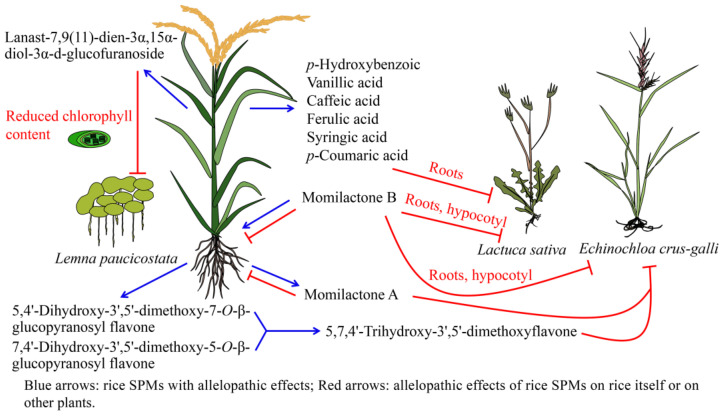
Allelopathic effects of SPMs from rice on other plants. Blue arrows: rice SPMs with allelopathic effects; red arrows: allelopathic effects of rice SPMs on rice itself or on other plants.

**Table 1 ijms-24-17053-t001:** Pathogens and herbivores of rice.

Category	No.	Family	Name	Site of Infection/Herbivory	Degree of Harm *
Herbivores	1	Delphacidae	*Nilaparvata lugens*	Xylem, phloem tissues	+++
	2		*Sogatella furcifera*	Xylem, phloem tissues	+++
	3		*Laodelphax striatellus*	Leaves, stem	+
	4	Miridae	*Trigonotylus caelestialium*	Shoots	++
	5	Pyralidae	*Chilo suppressalis*	Leaves	++
	6	Aleyrodidae	*Bemisia tabaci*	Leaves	++
	7	Lasiocampidae	*Malacosoma disstria*	Leaves	+
	8	Sphingidae	*Manduca sexta*	Leaves, stem	+
	9		*Mythimna loreyi*	Leaves	++
	10	Curculionidae	*Oryzophagus oryzae*	Leaves, stem	++
Pathogens	1	Agonomycetaceae	*Rhizoctonia solani*	Culms, sheath	+++
	2	Pyriculariaceae	*Magnaporthe grisea/Pyricularia grisea*	Leaves, nodes, stems, panicles, roots	+++
	3		*Magnaporthe oryzae/Pyricularia oryzae*	Leaves, nodes, stems, panicles, roots	+++
	4	Pleosporaceae	*Cochliobolus miyabeanus/Bipolaris oryzae*	Leaves	+++
	5	Dematiaceae	*Helminthosporium oryzae*	Grain	+++
	6	Agonomycetaceae	*Sclerotium rolfsii*	Stem	+
	7	Tuberculariaceae	*Fusarium proliferatum*	Spikelet	+++
	8	Hypocreales	*Sarocladium oryzae*	Leaf sheath	++
	9	Botryosphaeriaceae	*Macrophomina phaseolina*	Stem	++
	10	Burkholderiaceae	*Burkholderia glumae*	Husk	++
	11	Pseudomonadaceae	*Xanthomonas oryzae* pv. *oryzae* (*Xoo*)	Leaves	+++
	12		*X*. *oryzae* pv. *oryzicola* (*Xoc*)	Leaves	++
	13		*X. campestris* pv. *oryzae*	Leaves	+++

* +++: very serious; ++: moderately serious; +: serious.

## Data Availability

The Appendix A is available from the Online Library or from the author.

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
