# Peer review of "Chemical Structure Diversity and Extensive Biological Functions of Specialized Metabolites in Rice"

_ijms, 2023, doi:10.3390/ijms242317053_

Round 1
Reviewer 1 Report
Comments and Suggestions for Authors
Comments to the manuscript ijms-2713560 "Chemical structure diversity and extensive biological functions of specialized metabolites in rice".
Authors propose a review of the scientific literature on the chemical nature of the rice specialized metabolites and a critical presentation of their biological activities. The manuscript is well organized and carefully written. In my opinion, it represent a good contribute to the field of study. After some minor editing changes the manuscript is suitable for publication.
In order to improve the quality of the review, I have just two suggestions to the Authors.
1) Please include a short section of Methodology with a description of the details of the research: database used; years; key-words; total number of papers examined, etc.
2) Figures are very useful to better understand some of the scientific objectives of the review. However, due to the small size, there is a loss of quality and some details may be better evidenced increasing the size. I suggest to improve the readability of the Figures.
Author Response
1) Please include a short section of Methodology with a description of the details of the research: database used; years; key-words; total number of papers examined, etc.
Response: At the end of this review, 8.Materials and Methods are added as follows,
Two databases, SciFinder and web of science, were used to investigate the references from 1976 to 2023. The keywords were rice, rice specialized metabolism, biological functions, including the biosynthetic pathway and biological functions of terpenes, phenolics, and alkaloids, as well as the targets of these specialized metabolites on herbivores, and the detoxification effects of herbivores and fungi on specialized metabolites. 210 references were found, and 195 of them were cited.
2) Figures are very useful to better understand some of the scientific objectives of the review. However, due to the small size, there is a loss of quality and some details may be better evidenced increasing the size. I suggest to improve the readability of the Figures.
Response: Thank you for your timely reminder. All the figures have been re-checked, in order to increase the readability of the images and keep the displayed content consistent with the original content, some minor adjustments have been made to some contents, as follows, in figure 1, the herbivore promoting JA pathway has been modified. In figure 3, "Screen tube" is changed to "Sieve tube" in the paper and replace "Naringenin 7-O-β-D-xylopyranoside" with "Naringenin 7-O-β-D-xylopyranoside". In figure 4, "OsCYP76M8" is changed to "CYP76M8" in the review. The font size of all images has been adjusted.
The highlighted manuscript is in the attachment file.
Reviewer 2 Report
Comments and Suggestions for Authors
The paper is well written, and covers a topic which is not so easy to find in the literature. It fits into the scope of the journal.
There is only one thing that in my opinion is missing. Figures are very explicatives for the metabolic pathways, but the most interesting information for chemists, and people working with natural pesticides would be a table summarizing all the SPMs in rice which are efective against insects, with its formula and molecular structure. This would help readers interested in their isolation or in designing analogues.
I have read the whole manuscript, and in my opinion is an excellent review. Each section is appropriate and informative, the only thing I missed was some molecular data, in regard to the structure of the metabolites, which can be addressed including the figure I was suggesting.
The references are complete and appropriate.
And I did nor detect any issue on the supplementary data.
Author Response
The references are complete and appropriate.
And I did not detect any issue on the supplementary data.
Response: Thank you for your constructive comments. The chemical structures and names of specialized metabolites in rice have been summarized in the appendix. The names of herbivores and pathogenic bacteria and fungi, species, infection sites and infection degree in this review have all been summarized.
Reviewer 3 Report
Comments and Suggestions for Authors
The present manuscript provides a comprehensive overview of specialized metabolites (SPMs) in rice, detailing 439 compounds across various classes such as steroids, terpenoids, phenolic compounds, alkaloids, and others. The authors have precisely compiled and summarized the structural diversity and biological functions of these SPMs, emphasizing their integral roles in the plant's defence mechanisms against biotic stresses and allelopathic interactions.
While the topic of SPMs in rice is not novel, the depth of this review is commendable, offering a valuable reference source for researchers in the fields of plant biology, biochemistry ect.
The manuscript is well-structured and well written with a clear, logical flow that guides readers through. The authors have successfully made a vast amount of information accessible and informative, which is proved by summarizing the current state of knowledge.
The review identifies several knowledge gaps, particularly in the understanding of the biosynthesis and regulation of these metabolites, which could serve as fundamental points for future research directions. This aspect adds to the novelty of the manuscript.
The quality of the manuscript is high, with data and analyses from the literature presented in a clear and organized style. The references cited are recent and relevant, reflecting the current state of research.
The conclusions drawn are reasoned and well-supported by the literature, and the visual aids are appropriately used to enhance understanding.
In terms of scientific soundness, the review stands on the strength of the studies it references. The manuscript does not present new experimental data but relies on the quality and findings of previous research.
The review's content is within the subject of the journal, as it addresses a comprehensive range of compounds with significant implications for plant defence and agricultural practices. The potential applications in crop resistance and the rice industry underscore the manuscript's relevance and appeal to a broad audience.
Overall, this is a strong review that makes a valuable contribution, and I believe it is worthy of publication in your journal.
Author Response
Response: Thank you for your constructive comments, your affirmation and support will bring us the greatest encouragement.